# Sex Differences in the Influence of Sleep on Body Mass Index and Risk of Metabolic Syndrome in Middle-Aged Adults

**DOI:** 10.3390/healthcare8040561

**Published:** 2020-12-14

**Authors:** Hyun-E Yeom, Jungmin Lee

**Affiliations:** 1College of Nursing, Chungnam National University, Munhwaro 266, Junggu, Daejeon 35015, Korea; yeom@cnu.ac.kr; 2Youth Independence & Competencies Research Office, National Youth Policy Institute, Social Policy BLDG, Sejong National Research Complex, 370, Sicheong-daero, Sejong-si 30147, Korea

**Keywords:** body mass index, metabolic syndrome, sex characteristics, sleep disorders

## Abstract

Poor sleep and obesity are intimately related to cardiovascular diseases. We aimed to examine whether the influence of sleep and body mass index (BMI) on the risk of metabolic syndrome (MetS) differed by sex in middle-aged people. It is a cross-sectional study of 458 Korean participants who completed self-administered surveys; the data were analyzed using the PROCESS macro for SPSS. We found that both sleep and BMI were significant predictors of MetS risk in women, particularly by the role of BMI connecting the impact of sleep to MetS risk. However, the association was not found in men, showing that BMI, but not sleep, was a significant predictor of MetS. This sex-related difference was due to different relationships between sleep and BMI, indicating that BMI was more dependent on sleep quality for women than for men. Therefore, a sex-specific approach to decrease the risk of MetS is warranted.

## 1. Introduction

Cardiovascular diseases are one of the leading causes of increasing mortality and reduced quality of life globally. Metabolic syndrome (MetS), a condition clustered with cardiovascular risk factors (hypertension, central obesity, dyslipidemia in lower high-density lipoprotein (HDL), elevated triglycerides, and elevated fasting glucose), is a critical, life-threatening issue [1]. Recent data show that with increases in obesity in Asian countries, the prevalence of cardiovascular diseases and the risk of MetS have also increased. Asian populations are thought to have a relatively low risk of cardiovascular diseases [2]. However, a recent comparative study of Asian and European populations indicated the prevalence was similar to, or higher than, some less developed Asian countries than in Europe [3]. Also, according to recent South Korean national health statistics, the incidence of MetS was 26%, and 73.2% of adults who had received health screening services and had one or more MetS risk factors [4].

The prevalence of MetS is related to lifestyle factors, such as physical activity, diet, smoking, and drinking alcohol [5]. In particular, a large body of literature [6,7] indicates the key role of sleep in increasing the risks of MetS, showing intimate interrelatedness of sleep characteristics, such as duration, regular patterns, and overall quality, to risks of various cardiovascular diseases. Also, an increasing number of studies have provided a scientific mechanism for the link between sleep and obesity. Sleep affects energy balance and appetite-related hormones, such as ghrelin and leptin; these, in turn, are linked to physiological metabolic function. There is strong evidence that sleep deprivation is also associated with overall adiposity [6,7].

In addition, sleep and obesity are key contributors to the risk of MetS [5,6,7]. However, their interrelatedness and relative impact on MetS varies depending on sex, possibly due to differences in body composition. A comparative study of adolescents has reported sex-related differences in relationships between sleep, body mass index (BMI), and risk of obesity [8]. Some studies have shown that males’ body weight tends to be more sensitive to changes in sleep hours [8], while others have reported that females’ weight tends to be more sensitive to changes in sleep quality [9]. However, empirical evidence on sex-related differences in the relationship between sleep and BMI is inconclusive.

Midlife is accompanied by multifaceted aging-related declines in physiological and cardiometabolic functions that may cause sleep disturbance and increase BMI, which increases the risk of cardiovascular disorders and MetS [10]. Studies have shown that the risk of MetS increases with age, particularly after midlife in women with dramatic hormone declines with menopause (i.e., estrogen) [10,11]. Global epidemic evidence has shown that the prevalence of MetS is higher in men under 50 years old but higher in women over 60 [11]. Additionally, for women, sleep plays a pivotal role in cardiovascular health with increasing age, particularly after menopause; the relationship is not clear for men [12].

Thus, although numerous studies have shown the interrelationship among sleep, BMI, and risk of MetS, it remains unclear how it differs by sex in middle-aged people. The aims of this study were to examine sex-related differences in the relationship between sleep and BMI and to investigate whether the impacts of sleep and BMI on the risk of MetS differed by sex in middle-aged Korean people.

Figure 1 displays our hypothesized model. We hypothesized that (1) the relationship between sleep and BMI differs by sex, and (2) such a sex-related difference produces different impacts of sleep and body mass index on the risk of MetS in middle-aged men and women.

## 2. Materials and Methods

### 2.1. Participants and Procedures

In this cross-sectional study, we conducted a secondary analysis of data from two primary studies that aimed to investigate the characteristics of physiological and psychosocial experiences in midlife and perceptual and behavioral factors related to MetS risk factors in the Korean population.

Participants were recruited from multiple outpatient sites at two hospitals and two local clinics and community centers in the city of Daegu and the province of Gyeongbuk, South Korea. Data for 458 adults aged 45–64 years were collected through self-administered questionnaires. The exclusion criterion was individuals who had emergent health issues, such as stroke and coronary artery disease. 

Ethical considerations: Procedures for the two primary studies were approved by the Institutional Review Board of Chungnam National University (approval numbers: 201711-SB-081-01, 201809-SB-137-01). The study’s purposes and procedures, protection of participants’ rights, and measures for the physical and psychological well-being of human participants were explained to participants, along with the potential benefits and risks of participation. Participants provided written informed consent.

Sample size adequacy was validated by G * Power 3.1.9.2 [13]. For post-hoc testing under the conditions of type 1 error (alpha = 0.05), medium effective size (0.15) [14] for linear multiple regression analysis with 10 predictors and total sample size of 450, achieved power (1-β error probability) was 0.99.

### 2.2. Measures

#### 2.2.1. Risk of MetS

People with MetS have multiple risk factors for cardiovascular diseases, including type II diabetes, hypertension, and hyperlipidemia. The assessment of MetS risk was conducted via participant self-reporting on five dichotomous questions evaluating whether or not the participant met any of the five criteria announced by the National Cholesterol Education Program Expert Panel [15]: (1) high systolic (≥130 mm Hg) and/or diastolic (≥85 mm Hg) blood pressure or being treated with antihypertensive agents; (2) hyperglycemia (fasting blood glucose ≥ 100 mg/dL) or taking medication prescribed for diabetes; (3) hypertriglyceridemia (fasting plasma triglyceride ≥ 150 mg/dL) or taking medication prescribed for hyperlipidemia; (4) low high-density lipoprotein cholesterol (men < 40 mg/dL, women < 50 mg/dL) or taking medication prescribed for hyperlipidemia; and (5) central obesity, based on World Health Organization guidelines for Asian populations (abdominal circumference ≥90 cm in men, ≥85 cm in women) [16]. Participants were classified into the MetS risk group if they met two or more of the criteria.

#### 2.2.2. Sleep

Characteristics of sleep were assessed using the Korean version of the Pittsburgh Sleep Quality Index (PSQI), a valid global scale encompassing dimensions of subjective quality of sleep, latency, duration, and disturbances initiating and maintaining sleep; use of medication for sleep; and daytime dysfunction related to sleep [17]. The PSQI consists of 19 items on a 4-point Likert-type scale, ranging from 0 to 3. The quality of sleep score was calculated by an algorithm; higher scores indicate poorer quality of sleep. The reliability in this study was Cronbach’s alpha = 0.88.

#### 2.2.3. Socioeconomic and Health-Related Characteristics

As potential covariates, we assessed sociodemographic variables including age, sex, current job status, education, whole-family monthly income, and living status. For general health-related characteristics, an individual’s height and weight were self-reported and we calculated BMI using the formula (kg/m^2^). We also assessed engagement in healthy behaviors with the Health Behavior scale [18]. The scale is composed of 25 items that ask the frequency of health behaviors along several dimensions, including dietary habits (8 items), physical activities (4 items), stress management (5 items), health responsibility (5 items), and smoking (3 items). Responses are provided on a 4-point Likert scale from 1 (never) to 4 (routinely). The mean score of all items is calculated and a higher overall score means more active engagement in healthy lifestyles. The Health Behavior scale has been validated to assess the characteristics of healthy behavior to prevent cardiovascular diseases [18] and the reliability in this study was Cronbach’s alpha = 0.82.

### 2.3. Statistical Analyses

Data were analyzed using SPSS 26.0 (IBM Corp., Armonk, NY, USA). Sociodemographic and health-related characteristics were analyzed using descriptive statistics (frequency, percentage, mean, standard deviation. Bivariate correlation analysis was conducted to assess the relationship among study variables. For Pearson correlations, all categorical variables (i.e., level of education, living status, employment, level of family income, and MetS risk) were created as binary variables (like dummy variables) to interpret the finding as a point-biserial correlation coefficient. Finally, covariates were determined based on the correlation coefficients.

We hypothesized that the effect of quality of sleep on BMI may differ by sex, and the quality of sleep may affect MetS risk through BMI in a sex-dependent manner. The moderated mediation effects of quality of sleep, sex, and BMI in predicting MetS risk were tested using Hayes’ PROCESS macro for SPSS [19]. PROCESS applies a logistic regression when there is a dichotomous dependent variable (i.e., MetS risk), and conducts a series of logistic regressions to estimate direct and indirect effects and the proposed mediator path [19,20]. To test our proposed moderated mediation model (see Figure 1), the PROCESS macro first estimated an effect of sex moderating the relationship between quality of sleep and BMI, which was an interaction effect of sex and quality of sleep in predicting BMI. In the second step, it estimated an effect of BMI mediating the effect of quality of sleep on MetS risk; in other words, an indirect effect of quality of sleep on MetS risk through BMI. In the final step, it tested whether an indirect effect of the quality of sleep on MetS risk through BMI differed by sex, which would indicate a moderated mediation effect. Results of the three steps allowed us to determine whether there was a sex-moderated mediation effect of BMI in the relationship between quality of sleep and MetS risk.

## 3. Results

### 3.1. Descriptive Characteristics of Participants

Table 1 presents the sociodemographic information of the participants depending on sex. The number of participants in the MetS risk group was 226 (49.3%). Prevalence of MetS risk did not differ based on sex (*t* = 0.08, *p* = 0.780) and 52.2% of the MetS risk group were women. The average age of participants was 54.57 years (SD = 8.5 years). Approximately 64% were college graduates or beyond and most of the participants lived with family (92.4%) and were currently employed (78.2%). As for health-related characteristics, the average score of healthy behaviors in all participants was 2.57 (SD = 0.05) and did not differ by sex (*t* = 1.00, *p* = 0.317).

The mean levels of the main study variables were 6.16 (SD = 3.3) for quality of sleep and 24.25 (SD = 3.2) for BMI. The number (percent) of participants who were classified as overweight (BMI = 23–24.9 kg/m^2^), obese (BMI = 25–29.9 kg/m^2^), and severely obese (BMI ≥ 30 kg/m^2^) was 123 (26.9%), 151 (33.0%), and 20 (4.4%), respectively. Both quality of sleep (*t* = 2.61, *p* = 0.009) and BMI (*t* = −6.34, *p* < 0.001) did differ significantly by sex. Additional characteristics are presented in Table 1.

### 3.2. Correlations between the Main Study Variables and Sex-Related Differences

Bivariate correlation analysis showed that the risk of MetS was related to quality of sleep (*r* = 0.161, *p* < 0.001), BMI (*r* = 0.296, *p* < 0.001), age (*r* = 0.387, *p* < 0.001), level of education (*r* = −0.386, *p* < 0.001), living status (*r* = 0.094, *p* < 0.05), employment (*r* = −0.179, *p* < 0.001), and family income (*r* = −0.293, *p* < −0.001). Thus, these were included as covariates to assess the effect of sleep quality on BMI and MetS risk. 

Table 2 shows different correlational relationships among the main variables between men and women. For women, quality of sleep was significantly correlated with BMI (*r* = 0.165, *p* < 0.05) and MetS risk (*r* = 0.234, *p* < 0.001). Also, there was a significant correlation between BMI and MetS risk (*r* = 0.321, *p* < 0.001). For men, quality of sleep was not significantly related to BMI (*r* = −0.040, *p =* 0.552) or MetS risk (*r* = 0.069, *p =* 0.307). A significant correlation was found only for BMI and MetS risk (*r* = 0.305, *p* < 0.001).

### 3.3. Sex Difference in Effects of Quality of Sleep on BMI and MetS Risk

The PROCESS macro model 7 results for the effect of sex on the relationship between quality of sleep, BMI, and risk of MetS are as follows (see Table 3). First, a significant interaction between quality of sleep and sex (B = −0.21, t = −2.24, *p* < 0.05) was found in predicting BMI. It indicates that the relationship between quality of sleep and BMI differed by sex and sex moderated the effect of quality of sleep on BMI. Figure 2 displays the relationship between sleep quality and BMI by sex; for women, worse sleep quality predicted higher BMI, and for men, the reverse.

Second, the indirect effect of quality of sleep on the risk of MetS through BMI was significant in women (indirect effect = 0.04, 95% CI = 0.001, 0.083) but not in men (indirect effect = −0.03, 95% CI = −0.076, 0.019). It indicates a difference in indirect effects of quality of sleep on MetS risk through BMI between women and men.

Finally, the result for the whole model 7 indicated that sex moderated the effect of BMI, mediating the relationship between quality of sleep and risk of MetS, confirming a moderated mediation effect (Index = −0.07, 95% CI = −0.13, −0.01).

## 4. Discussion

Despite extensive evidence of interrelatedness among sleep, obesity, and MetS, the information about sex differences in the impacts of sleep and obesity on the risk of MetS is inconsistent. The findings of the current study describe the sophisticated mechanism by which sex changes the ways sleep and BMI contribute to increased risks for MetS. The sex-related differences in the relationship between sleep and BMI account for variances in how sex impacts increased risk of MetS in middle-aged people.

A key finding is that quality of sleep is a primary factor affecting the risk of MetS through the joint impact with BMI, but with a difference by sex. The women in midlife who have poorer quality of sleep tended to have higher BMI, leading to an increase in the risk of MetS. Notably, BMI played a role as a mediator for the relationship between sleep and the risk of MetS. However, such an association was not found in midlife men. The findings indicate that deterioration of quality of sleep and an increase of BMI need to be considered relevant predictors for the risk of MetS in women. Comparably, sleep in men was a relatively less influential factor for BMI; therefore, BMI rather than sleep needs to be considered a relatively more meaningful factor related to the risk of MetS.

It is noteworthy that the sex-dependent distinguished impacts of sleep and BMI on MetS risk were associated with the different connections of sleep to BMI, comparable by sex. Women’s BMI tended to be intimately related to the quality of sleep; this tendency was not detected in men. This finding conflicts with prior evidence that sleep was closely related to BMI in both men and women [3]. Some studies have addressed that sex-related physiological distinctiveness may cause such a difference [9]. A recent study [21] reported a significant relationship between sleep disorders and obesity in postmenopausal women, and another study [22] addressed the changes in women’s sleep with regard to sex hormone-related changes beginning from the perimenopausal period. Participants in this study were middle-aged, and it is known that women tend to experience more turbulent physiological changes during the menopause period. Therefore, further study is necessary to include physiological characteristics for a more concrete understanding of the sex-related differences in the relationship between sleep and BMI. It is also notable that the current study added empirical evidence that the impacts of sleep and BMI on MetS risk differed by sex, distinguishing the relationship between sleep and BMI depending on sex. This study found that despite strong evidence for the impact of sleep on MetS, its contribution to increasing the risk of MetS differed according to sex when considering its interrelatedness with BMI, since BMI was associated with sleep only in women.

Another interesting finding included confirmation of additional sex-related differences in BMI changes by age. BMI tended to increase with age for women, whereas this tendency was reversed for men. The findings are consistent with previous studies reporting that women have a high tendency to gain weight with increasing age, mainly due to menopause-related changes including body composition and cardiometabolic risks [10]. Moreover, consistent with previous studies [23,24] we found that the quality of sleep in women tends to be worse as age increases, while this tendency was not observed in men. This finding does conflict with some evidence that men rather than women tended to suffer worse sleep deterioration with increased age [25] and the quality of sleep tended to decrease as age increased in middle-aged (46–60 years old) men [24]. Therefore, the sex-related difference in the relationship between sleep and aging is still inconclusive.

On the other hand, consistent with strong previous evidence, the risk of MetS increases with age in both men and women. The finding emphasizes the importance of efforts to prevent various MetS risks at a younger age. Middle-aged adults’ physiological characteristics are unstable and turbulent, depending on individualized aging-related declines, and they may show substantial variation even within this sexual homogeneity. Therefore, further study is warranted to confirm and clarify the sex-related differences.

The other considerable finding is the interrelatedness between sleep and lifestyles in men and women. We found that individuals of both sexes who more actively engage in healthy lifestyle practices, such as physical exercise and diet, tended to report better quality of sleep. This finding is congruent with existing evidence on the benefits of health behavior for better sleep [26]. However, a noticeable relationship between healthy lifestyle practices and BMI was not found, despite considerable evidence supporting these links [27]. Prior research has indicated the necessity of healthcare behavioral strategies targeting specific health outcomes [28]. Therefore, future studies focusing on the link between the risk of MetS and healthy behavior by type are suggested.

### Strengths and Limitations

A major contribution of the current study is that it clearly indicated that for women, but not men, quality of sleep has a substantial impact on risk of MetS with increased BMI, due to the intimate connection between sleep quality and BMI. Therefore, the present study provides a sex-differentiated concrete association among sleep, BMI, and the risk of MetS; in particular, a different role of BMI by sex in the relationship between sleep and the risk of MetS.

Several limitations of this study need to be addressed. First is its cross-sectional design, which aimed to confirm the causal effect of sleep and BMI on MetS. Further longitudinal studies are necessary for more convincing, predictive evidence of this relationship. Additionally, the population of this study was ethnically homogeneous. Empirical studies have shown that the prevalence of MetS varies by ethnicity. A recent study [3] that compared Asian and European populations reported differences in the prevalence of MetS and risk factors by sex and ethnicity; its prevalence was higher in women than in men among the Asian population, while the tendency was reversed in the European population. In addition, factors contributing to MetS were inconsistent, although hypertension was the most common risk factor for MetS in both populations. Furthermore, a global report [29] has shown a considerable variation in the prevalence of risk factors of MetS even within Asia-Pacific countries, with a much higher rate in low-middle- and low-income countries compared to developed countries. Therefore, further cross-cultural studies are necessary for an integrated understanding of the behavioral and bio-social factors related to the risk of MetS. Also, the current study employed BMI rather than abdominal obesity. Some prior studies have indicated a need to distinguish overall obesity and abdominal obesity based on differences in their relationships with MetS [3]. For both Asian (Indonesian) and European (Dutch) populations, the prevalence of MetS had a stronger relationship with abdominal circumference than with BMI (and thus with abdominal adiposity than overall adiposity) [3]. Thus, a further study focusing on abdominal obesity’s contribution to the risk of MetS, with consideration of sex-related differences, is suggested. 

## 5. Conclusions

The current study highlights the need to consider sex-related differences to reduce the risk of metabolic syndrome by demonstrating that, for women, both sleep quality and BMI are influential predictive factors for the risk of MetS, whereas for men, BMI but not the quality of sleep is a critical issue. The differences in the roles of BMI were based on different relationships between sleep and BMI by sex. Sex-specific strategies that target the prevention of MetS and effective management of its risk factors are highly warranted.

## Figures and Tables

**Figure 1 healthcare-08-00561-f001:**
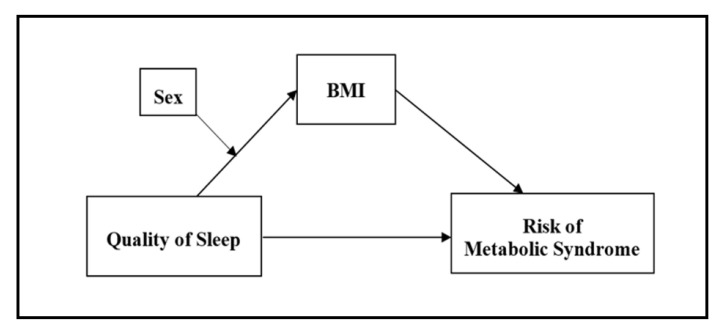
Hypothesized sex-moderated mediation model for the relationship between quality of sleep, body mass index (BMI), and risk of metabolic syndrome (MetS).

**Figure 2 healthcare-08-00561-f002:**
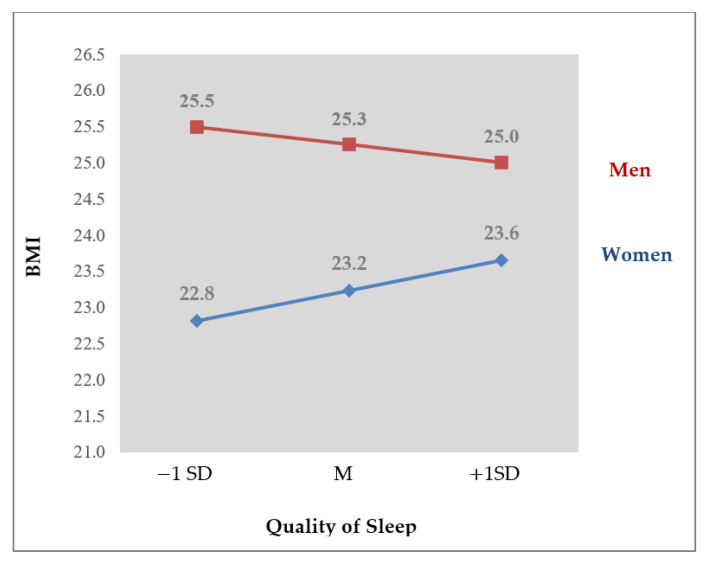
Sex differences in the relationship between worsening of quality of sleep and BMI.

**Table 1 healthcare-08-00561-t001:** General characteristics of the participants and sex-related differences (*N* = 458).

Variables	Categories	All	Sex		Differences by Sex
			Women (*n* = 236)	Men (*n* = 222)	
		M ± SD or n (%)	M ± SD or n (%)	M ± SD or n (%)	*t* or χ^2^ (*p*)
Age		54.57 ± 8.5	55.81 ± 9.1	53.24 ± 7.7	3.27 (0.001)
High education	Yes	291 (63.5)	113 (47.9)	178 (80.2)	51.51 (0.000)
(≥College)	No	167 (36.5)	123 (52.1)	44(19.8)	
Living alone	Yes	35 (7.6)	23 (9.7)	12 (5.4)	3.05 (0.081)
	No	423(92.4)	213 (90.3)	210 (94.6)	
Currently employed	Yes	358 (78.2)	153 (64.8)	205 (92.3)	52.54 (0.000)
	No	96 (20.9)	81 (34.3)	15 (6.8)	
	Missing	4 (0.9)	2 (0.9)	2 (0.9)	
High family income(≥4,000,000 won/mon)	Yes	239 (52.2)	89 (37.7)	120 (54.1)	12.02 (0.001)
	No	209 (45.6)	141 (59.8)	98 (44.1)	
	Missing	10 (2.2)	6 (2.5)	4 (1.8)	
Health behaviors		2.57 ± 0.5	2.6 ± 0.5	2.5 ± 0.5	1.00 (0.317)
Body mass index		24.25 ± 3.2	23.4 ± 2.8	25.2 ± 3.3	−6.34 (0.000)
Quality of sleep		6.16 ± 3.3	6.5 ± 3.4	5.8 ± 3.0	2.61 (0.009)
Risk of Metabolic Syndrome	Yes	226 (49.3)	118 (50.0)	108 (48.6)	0.08 (0.780)
	No	232 (50.7)	118 (50.0)	114 (51.4)	

**Table 2 healthcare-08-00561-t002:** Correlations between demographic and main variables by sex (*N* = 458).

Variables		Men (*n* = 222)								
		Age	Education ^§1^	Living Status ^§2^	Employment ^§3^	Family Income ^§4^	Health Behaviors	Quality of Sleep	BMI	Mets Risk ^§5^
**Women** **(n = 236)**	**Age**	-	−0.046 **	−0.060	−0.438 **	−0.186 **	0.256 **	0.125	−0.223 **	0.220 **
	**Education ^§1^**	−0.534 **	-	−0.231 **	0.367 **	0.438 **	0.091	−0.186 **	0.018	−0.375 **
	**Living status ^§2^**	0.335 **	−0.201 **	-	−0.253 **	−0.146 *	−0.044	0.154 *	−0.039	0.006
	**Employment ^§3^**	−0.295 **	0.272 **	−0.152 **	-	0.242 **	0.001	−0.196 **	0.024	−0.242 **
	**Family income ^§4^**	−0.525 **	0.487 **	−0.228 **	0.239 **	-	0.048	−0.043	−0.059	−0.226 **
	**Health behaviors**	0.169 **	−0.103	−0.057	−0.104	0.093	-	−0.245 **	−0.046	−0.026
	**Quality of sleep**	0.205 **	−0.228 **	0.059	−0.024	−0.156 *	−0.149 *	-	−0.040	0.069
	**BMI**	0.163 *	−0.207 **	0.072	−0.074	−0.084	−0.111	0.165 *	-	0.305 **
	**Mets risk ^§5^**	0.527 **	−0.433 **	0.157 **	−0.171 **	−0.363 **	0.053	0.234 **	0.321 **	-

Note. ^§^ Dummy variable; Reference groups: ^1^ = high education (≥College); ^2^ = living alone; ^3^ = currently employed; ^4^ = high family income (≥4,000,000 won/mon); ^5^ = MetS risk group. * *p* < 0.05, ** *p* < 0.001.

**Table 3 healthcare-08-00561-t003:** Sex-related different effects of quality of sleep on BMI and MetS risk (n = 439).

Predictor	BMI			
	B	SE	*t/z*	*p*
Quality of sleep	0.12	0.06	1.98	0.05
Sex ^§,1^	3.40	0.66	5.16	0.00
Quality of sleep × sex ^§,1^	−0.21	0.09	−2.24	0.03
Covariates				
Constant	26.26	1.53	17.18	0.00
Age	−0.04	0.02	−1.80	0.07
Education ^§,2^	−0.87	0.41	−2.09	0.04
Living alone ^§,3^	0.03	0.58	0.06	0.95
Employment **^§,4^**	−0.26	0.42	−0.61	0.54
Family income **^§,5^**	−0.14	0.34	−0.40	0.69
Health behaviors	−0.34	0.32	−1.06	0.29
R^2^ = 11.29%, F (9, 429) = 6.07, *p* < 0.000				
**Predictor**	**MetS risk**			
	**B**	**SE**	***t/z***	***p***
BMI	0.31	0.04	7.00	0.00
Quality of sleep	0.03	0.04	0.73	0.46
Covariates				
Constant	−12.07	1.80	−6.72	0.00
Age	0.10	0.02	5.19	0.00
Education ^§,2^	−0.94	0.31	−3.06	0.00
Living alone ^§,3^	−0.32	0.49	−0.65	0.51
Employment **^§,4^**	−0.01	0.34	−0.02	0.98
Family income **^§,5^**	−0.65	0.26	−2.49	0.01
Health behaviors	−0.05	0.26	−0.19	0.85
**Logistic regression summary** McFadden’s R^2^= 26.83, CoxSnell’s R^2^=31.03, Nagelkrk’s R^2^ = 41.40 F (8, 430) = 163.11, *p* < 0.000				
**Sex difference in indirect effects of quality of sleep on MetS risk BMI**				
Sex ^§,1^	Boot indirect effect	Boot SE	LL 95% CI	UL 95% CI
Men	−0.03	0.02	−0.076	0.019
Women	0.04	0.02	0.001	0.083
**Index of sex-moderated mediation effect for the relationship** **between quality of sleep, BMI, and MetS risk**				
	Index	Boot SE	LL 95% CI	UL 95% CI
BMI	−0.07	0.03	−0.13	−0.01

Note. MetS = metabolic syndrome. ^§^ Dummy variable; Reference groups: ^1^ = Men; ^2^ = high education (≥College); ^3^ = living alone; ^4^ = currently employed; ^5^ = high family income (≥4,000,000 won/mon).

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
