# Peer review of "Sex Differences in the Influence of Sleep on Body Mass Index and Risk of Metabolic Syndrome in Middle-Aged Adults"

_healthcare, 2020, doi:10.3390/healthcare8040561_

Round 1

Reviewer 1 Report

The methods of analysis and description of methods raise the most reservations. The main variable is dichotomous (1-fulfilment of the MetS criterion). First of all, it is not described in the results how many people met this criterion (total, men, women?). Secondly, Hayes' method concerns linear regression. In case of the dichotomous dependent variable logistic regression should be used. The authors also count the Pearson correlation  including nominal and dichotomous variables in the table, with MetS variable in the last column.          

The description of methods does not provide information about the use of linear regression. It appears when the sample size is justified. Using the term Macro Process is imprecise and a little colloquial. This is so called syntax for SPSS and there are thousands of them. It should be stated in the methods that it is a method of moderated mediation (conditional process). Information about PROCESS macro should not be included in the summary. It is better to write about moderated mediation. This term appears in the title of the figure with a model describing the hypothesis adopted in the paper, and it is not in the description of methods. The interpretation of the index given at the bottom of Table 3 requires an explanation for the less familiar reader, as these are very small values.

In line 72 the term cross-sectional correlational study combines approaches from the medical and social sciences. You can delete the term correlational or change it to observational because it is a confusion of classifications of epidemiological and social studies.   

The term high cholesterol in line 93 seems to be colloquial in comparison with further more clinical description.

In the line 112 please indicate whether height and weight were self-reported  or coming from measurements. The percentage of overweight and obese people was not given.

The Helath Behaviour scale is also very poorly described. Studying the quoted references it seems to be an extension of the scale from Song&Lee (2001). Previously 20 items were given, now it is 25.

The variable “having a job” is questionable and associated with unemployment, difficult to interpret when entering retirement age. Do people about 64 years old do not have pension rights in Korea for both sexes. Better to change into employment status.

Some of the variables in Table 2 are not described at all in methods like general health status.

The Sleep Quality scale has an unfortunate name if high values are negative. It is difficult to interpret relationships. It should be called Sleep Problems, or in similar way. 

The description of results starts with a description of the sample and correlations. Before correlations the main dependent variable should be described.

I think that the paper requires thorough rewriting, including a more accurate description of tools, methods and results. The methods should also be customized to the type of variables.  Perhaps unexpected results (decrease of BMI along with deterioration of sleep quality in women) result from methodological flaws.  

Author Response

Thank you for your reviews of the manuscript entitled “Sex Differences in the Influence of Sleep on Body Mass Index and Risk of Metabolic Syndrome in Middle-Aged Adults.” We appreciate the helpful feedback from the reviewers. We carefully addressed the reviewers’ comments and suggestions and incorporated them in the revised version of our paper. For your attention, we present the reviewers’ comments and provide our responses in the order in which they appeared in the reviews.

Point-by-point responses to Reviewer 1

  1. The methods of analysis and description of methods raise the most reservations. The main variable is dichotomous (1-fulfilment of the MetS criterion).
  • First of all, it is not described in the results how many people met this criterion (total, men, women?)

(Response) The information was present in Table 1 (“Yes” for the risk for metabolic syndrome: n=226). For clarification, we addressed the frequency in section 3.1 in the Results section (lines 152–154).

  • Secondly, Hayes' method concerns linear regression. In case of the dichotomous dependent variable logistic regression should be used.

(Response) Logistic regression is applied when a dependent variable is dichotomous. Hayes’ PROCESS macro can be used to conduct a logistic regression analysis. We added the reference for this in the paper.

Explanation of PROCESS: “If PROCESS detects only two distinct values on the variable listed in the y = specification, the direct and indirect effects as well as the path(s) from the proposed mediator(s) to the dependent variable are estimated using logistic regression.” From: Hayes, A.F. PROCESS: A versatile computational tool for observed variable mediation, moderation, and conditional process modeling [White paper]. 2012. Retrieved from http://www.afhayes.com/public/process2012.pdf. Additional citation for PROCESS: Hayes, A.F.; Matthes, J. Computational procedures for probing interactions in OLS and logistic regression: SPSS and SAS implementations. Behav Res Methods 2009, 41, 924–936.

  • The authors also count the Pearson correlation including nominal and dichotomous variables in the table, with MetS variable in the last column.          

(Response) We created binary variables (like dummy variables) for dichotomous categorical variables and conducted a Pearson correlation. This correlation is also known as a point-biserial correlation coefficient.

  1. The description of methods does not provide information about the use of linear regression. It appears when the sample size is justified. Using the term Macro Process is imprecise and a little colloquial. This is so called syntax for SPSS and there are thousands of them. It should be stated in the methods that it is a method of moderated mediation (conditional process). Information about PROCESS macro should not be included in the summary. It is better to write about moderated mediation. This term appears in the title of the figure with a model describing the hypothesis adopted in the paper, and it is not in the description of methods. The interpretation of the index given at the bottom of Table 3 requires an explanation for the less familiar reader, as these are very small values.

(Response) Thank you for your comment. As you suggested, we added more details in section 2.3 (Statistical Analysis) of the Methods section about this step-by-step approach (lines 136–148). We also revised the notes of Table 3.

  1. In line 72 the term cross-sectional correlational study combines approaches from the medical and social sciences. You can delete the term correlational or change it to observational because it is a confusion of classifications of epidemiological and social studies.   

(Response) Thank you for the comment. As suggested, we deleted the term “correlational” to avoid any confusion.

  1. The term high cholesterol in line 93 seems to be colloquial in comparison with further more clinical description.

(Response) We changed the term to “hyperlipidemia.”

  1. In the line 112 please indicate whether height and weight were self-reported or coming from measurements. The percentage of overweight and obese people was not given.

(Response) We added text in the Methods section describing that the height and weight information was from participant self-report (lines 117–119). The prevalence of overweight and obesity were added in section 3.1 of the Results section (lines 160–162).

  1. The Health Behaviour scale is also very poorly described. Studying the quoted references it seems to be an extension of the scale from Song &Lee (2001). Previously 20 items were given, now it is 25.

(Response) Song et al. (2004) extended items on a scale initially developed by Song & Lee (2000) and reported the reliability for the scale with 25 items. We added more information about the scale in the Methods section (lines 119–126).

  1. The variable “having a job” is questionable and associated with unemployment, difficult to interpret when entering retirement age. Do people about 64 years old do not have pension rights in Korea for both sexes. Better to change into employment status.

(Response) We changed “having a job” to “currently employed.” Thank you for the comment.

  1. Some of the variables in Table 2 are not described at all in methods like general health status.

(Response) We fixed Tables 1 and 2 and revised the text in the Methods and Results sections. We apologize for the mistakes.

  1. The Sleep Quality scale has an unfortunate name if high values are negative. It is difficult to interpret relationships. It should be called Sleep Problems, or in similar way. 

(Response) We understand what your concerns are. However, the Pittsburgh Sleep Quality Index (PSQI) is a valid scale that has been globally used to investigate the quality of sleep and encompasses several dimensions including not only problems, such as disturbances initiating and maintaining sleep, but also sleep duration, efficiency, and latency. Therefore, we think it is better to maintain the term “quality of sleep” to enable comparison with further studies on quality of sleep.

  1. The description of results starts with a description of the sample and correlations. Before correlations the main dependent variable should be described.

(Response) Thank you for the suggestion. We described the information of main dependent variable (i.e., risk of metabolic syndrome) in 3.1. Results section (line 152-154), which is the first part of Results section.  

  1. I think that the paper requires thorough rewriting, including a more accurate description of tools, methods and results. The methods should also be customized to the type of variables.  Perhaps unexpected results (decrease of BMI along with deterioration of sleep quality in women) result from methodological flaws.  

(Response) We did our best to provide additional details about the measures and statistical analyses in the Methods section. Also, we rechecked the adequacy of the type of variables. We did double check the analytic process and the interpretation of the results from the statistical analysis. We may need to clarify the interpretation of the finding regarding the relationship between quality of sleep and BMI in women. A higher quality of sleep score means a poorer quality of sleep. Thus, for women, the deterioration of quality of sleep was linked to an increased BMI. This relationship was reversed in men. Therefore, there was a sex difference in the relationship between quality of sleep and BMI.

Reviewer 2 Report

Line 77-78

The authors performed self-administered questionnaires for analysis; however, selection-bias of better educational background and may had more insight of healthy problems. I suggest to adjust education level in the statistic model. Besides, how could authors identify the metabolic syndrome through questionnaires?

Line 86-89

The number of collecting samples of 450 could achieve power at 0.99. However, the effect size (0.15) should be defined clearly. Are there any references to get the number?.

Line 119-124

The author should give more detail about Hayes’ PROCESS macro because it is an macro language instead of statistic methods.

Table1

The result of having job or not was not the only etiology of quality of sleep. There was shift system during work which will change the pattern of sleep. There was no data with mention the ratio of menopause participants which will later be an explanation of discussion

Line 138-140 and Table 2

The correlation of each factors of all population only showed in the paragraph but not in the Table 2. I suggest authors should integrate the results of all population, men, and women in the same table. Besides, what does M(SD) of income (4.21 in men and 3.44 in women) mean?

Line 188-192

Different result from men and women is noted in the study. As authors mention, menopause period might be important factor in this period. But authors should present more evidences or reference to support this assumption. If not, authors should mention in the limitation.

Line 202-203

The author found that the quality of sleep in women tends to be worse as age increases, while this tendency is not observed in men. But this conclusion is a result from table 2 with only Pearson’s correlation test. There might be confounding factors which lead to different results. The author should do regression test to confirm this conclusion.

Author Response

Thank you for your reviews of the manuscript entitled “Sex Differences in the Influence of Sleep on Body Mass Index and Risk of Metabolic Syndrome in Middle-Aged Adults.” We appreciate the helpful feedback from the reviewers. We carefully addressed the reviewers’ comments and suggestions and incorporated them in the revised version of our paper. For your attention, we present the reviewers’ comments and provide our responses in the order in which they appeared in the reviews.

Point-by-point responses to Reviewer 2

  1. Line 77-78: The authors performed self-administered questionnaires for analysis; however, selection-bias of better educational background and may had more insight of healthy problems. I suggest to adjust education level in the statistic model. Besides, how could authors identify the metabolic syndrome through questionnaires?

(Response) Thank you for the suggestion. We included education level as a covariate and added the information to Table 3.

For the assessment of risks of metabolic syndrome, the participants answered (self-report) five dichotomous questions. Participants checked a box (yes or no) to indicate whether or not they had any of the following items: (1) high systolic (≥ 130 mmHg) and/or diastolic (≥ 85 mmHg) blood pressure or being treated with antihypertensive agents; (2) hyperglycemia (fasting blood glucose ≥ 100 mg/dL) or taking medication prescribed for diabetes; (3) hypertriglyceridemia (fasting plasma triglyceride ≥ 150 mg/dL) or taking medication prescribed for hyperlipidemia; (4) low high-density lipoprotein cholesterol (men < 40 mg/dL, women < 50 mg/dL) or taking medication prescribed for hyperlipidemia; and (5) abdominal circumference ≥ 90 cm in men, ≥ 85 cm in women. We added this information in the Methods section (lines 94–105).

  1. Line 86-89: The number of collecting samples of 450 could achieve power at 0.99. However, the effect size (0.15) should be defined clearly. Are there any references to get the number?.

 (Response) From Cohen’s (1988) measure of the effect size in multiple regression, f2 is calculated as follows: (Please see the attached file)

We adopted the approach from Cohen (1988) that defined values near 0.02 as small, near 0.15 as medium, and above 0.35 as large.

  1. Line 119-124: The author should give more detail about Hayes’ PROCESS macro because it is a macro language instead of statistic methods.

(Response) We added more detail about Hayes’ PROCESS macro (lines 136–148) and changed some of the terms to more common language to aid the reader’s understanding.

  1. Table1: The result of having job or not was not the only etiology of quality of sleep. There was shift system during work which will change the pattern of sleep. There was no data with mention the ratio of menopause participants which will later be an explanation of discussion

(Response) Thank you for your important comments and we agree with your points. We addressed the issues in the Discussion section because we had no data regarding shift work or menopausal status (lines 230–232).

  1. Line 138-140 and Table 2: The correlation of each factors of all population only showed in the paragraph but not in the Table 2. I suggest authors should integrate the results of all population, men, and women in the same table. Besides, what does M(SD) of income (4.21 in men and 3.44 in women) mean?

(Response) Thank you for the comments. We changed Tables 1 and 2 as suggested.

  1. Line 188-192: Different result from men and women is noted in the study. As authors mention, menopause period might be important factor in this period. But authors should present more evidences or reference to support this assumption. If not, authors should mention in the limitation.

(Response) We appreciate your comment. We included more information regarding menopause-related effects along with references (lines 230–232, 245–247).

  1. Line 202-203: The author found that the quality of sleep in women tends to be worse as age increases, while this tendency is not observed in men. But this conclusion is a result from table 2 with only Pearson’s correlation test. There might be confounding factors which lead to different results. The author should do regression test to confirm this conclusion.

(Response) We understand your concern in that there are limitations in concluding causal relationships from correlational coefficients. In this study, quality of sleep was considered as an independent variable along with sex, and risk of metabolic syndrome was a dependent variable. We think that testing the relationship between quality of sleep and age (both were dependent variables) depending on sex using linear regression analyses may be misleading a main point of this study. Therefore, we tried to address the relationship between age and quality of sleep cautiously to not overestimate the relationship. We added more about this in the Discussion, along with references (lines 245–247).

Round 2

Reviewer 1 Report

The authors have improved the manuscript as suggested in the review. Still, however, Figure 2 may mislead the reader. High values of the quality of sleep scale mean more problems. The word quality itself, however, raises positive associations. I propose the title: Sex differences in the relationship between worsening of quality of sleep and BMI.   In this title there is no name for the scale but a general concept to which it refers.   Without such an explanation, the opposite association can be concluded.

Author Response

Thank you for your reviews of the manuscript entitled “Sex Differences in the Influence of Sleep on Body Mass Index and Risk of Metabolic Syndrome in Middle-Aged Adults.” We appreciate the helpful feedback from the reviewers. We carefully addressed the reviewers’ comments and suggestions and incorporated them in the revised version of our paper. For your attention, we present the reviewers’ comments and provide our responses in the order in which they appeared in the reviews.

Point-by-point responses to Reviewer 1

  • The authors have improved the manuscript as suggested in the review. Still, however, Figure 2 may mislead the reader. High values of the quality of sleep scale mean more problems. The word quality itself, however, raises positive associations. I propose the title: Sex differences in the relationship between worsening of quality of sleep and BMI.   In this title there is no name for the scale but a general concept to which it refers.   Without such an explanation, the opposite association can be concluded.

(Response) Thank you for your careful review and helpful suggestion. We changed the title of Figure 2 to “Sex differences in the relationship between worsening of quality of sleep and BMI,” as suggested.

Reviewer 2 Report

No further comments

Author Response

Thank you for your reviews of the manuscript entitled “Sex Differences in the Influence of Sleep on Body Mass Index and Risk of Metabolic Syndrome in Middle-Aged Adults.”

No further comments from reviewer 2.